# Facile Fabrication of High-Performance Composite Films Comprising Polyvinyl Alcohol as Matrix and Phenolic Tree Extracts

**DOI:** 10.3390/polym15061424

**Published:** 2023-03-13

**Authors:** Ying Xu, Bowen Liu, Lulu Zheng, Yunxia Zhou, Hisham Essawy, Xinyi Chen, Xiaojian Zhou, Jun Zhang

**Affiliations:** 1Yunnan Provincial Key Laboratory of Wood Adhesives and Glued Products, Southwest Forestry University, Kunming 650224, China; 2Department of Polymers and Pigments, National Research Centre, Dokki, Cairo 12622, Egypt

**Keywords:** tannin, lignin, polyvinyl alcohol, tensile strength, biodegradability

## Abstract

Given that tree extracts such as tannin and lignin can be efficiently used as modifying materials, this helps to verify the global trend of energy saving and environment protection. Thus, bio-based biodegradable composite film incorporating tannin and lignin as additives, together with polyvinyl alcohol (PVOH) as a matrix (denoted TLP), was prepared. Its easy preparation process endows it with high industrial value in comparison to some bio-based films with complex preparation process such as cellulose-based films. Furthermore, imaging with scanning electron microscopy (SEM) shows that the surface of tannin- and lignin-modified polyvinyl alcohol film was smooth, free of pores or cracks. Moreover, the addition of lignin and tannin improved the tensile strength of the film, which reached 31.3 MPa as indicated by mechanical characterization. This was accounted for by using Fourier transform infrared (FTIR) and electrospray ionization mass (ESI-MS) spectroscopy, which showed that the physical blending of lignin and tannin with PVOH was accompanied by chemical interactions that gave rise to weakening of the prevailing hydrogen bonding in PVOH film. In consequence, the addition of tannin and lignin acquired the composite film good resistance against the light in the ultraviolet and visible range (UV-_VL_). Furthermore, the film exhibited biodegradability with a mass loss about 4.22% when contaminated with *Penicillium* sp. for 12 days.

## 1. Introduction

Films have been widely used in electrical [1,2], machinery [3,4], printing [5,6] and other industries. As an environmentally friendly biodegradable film, PVOH film has become a hotspot due to its high transparency, good oil resistance, corrosion resistance and tear resistance [7,8]. However, the structural units of PVOH contain a large number of hydrophilic hydroxyl groups, which leads to poor water resistance, weak UV-_VL_ resistance, low hardness, low tensile strength, and easy adhesion after moisture absorption [9]. Some studies have been carried out successively to modify PVOH films in order to improve the solid content, hardness, water resistance and mechanical properties of the films. For example, starch-PVOH composite films were prepared by grafting oxidation of starch and oleic acid [10]; nevertheless the elongation at break of the films was only 42.67% due to the poor fluidity of starch. Likewise, PVOH film was modified with nanocellulose, however the component became turbid after cooling to form aging gel, resulting in poor brittleness and flexibility of the film [11]. In recent years, the problems of energy shortage and environmental pollution have become increasingly prominent. It is therefore of great significance to explore novel methodologies for environmental protection, likely by focusing on the use of renewable materials from agriculture and forestry. The use of these materials as modifiers in biodegradable PVOH films preparation can help to alleviate the energy shortage and achieve the target of low-carbon environment. Trees as forest resources are one of the most potential biomass materials. Further, wood is the main building material. In addition, condensed tannins, mainly extracted from bark, are water soluble extracts from plants [12]. They contain stable phenolic ring structure, which is similar to phenol, but exhibit higher reactivity than phenol [13,14]. The good thermal stability and rigidity of tannin resin render it frequently used in preparation of thermosetting biomass based polymeric materials, such as foam [15,16,17], adhesive [18,19,20], and grinding wheel [21]. Based on the above facts, it can be considered that blending condensed tannin with PVOH can improve the mechanical properties and thermal stability of PVOH. It is expected that adding tannin into PVA system can improve the tensile strength and hardness of the resulting film. However, due to the large steric hindrance, condensed tannins have poor reactivity with PVOH [22], and due to the large molecular weight of condensed tannins, they undergo agglomeration easily in PVOH, resulting in poor compatibility [23]. Therefore, in the case of the tannin and PVA system, a matrix with good adhesion properties is needed to uniformly fix the tannin in the PVA system during the mixing process. It is worth noting that lignin extracted from trees is the main by-product of the pulp and paper industry. It is rich in agricultural and forest resources, cheap, renewable, non-toxic, and has good adhesion properties [24,25]. However, due to low reactivity, most of the remaining lignin from paper industry is used for incineration or discarding, and only a small part is used to produce low value products [26,27]. Lignin has the function of bonding cellulose and hemicellulose in wood. It is a natural binding agent [28,29]. Therefore, it is assumed that lignin can be bonded with tannin and PVOH to further improve of the mechanical properties, thermal properties and UV-_VL_ absorption resistance of the films. Tannin and lignin are both tree resources. Blending them with PVA to prepare UV-resistant, high-strength film is beneficial to improve the utilization rate of wood processing residues. Hence, based on these facts together, we decided to go further in this study to deal with blending PVOH with tree extract of condensed tannin and lignin in presence of glycerol as a toughening agent to prepare bio-based film with good mechanical properties, thermal stability and UV-_VL_ absorption resistance. We wish that the resulting degradable film based on tree extract can expand the application of tannin and lignin in industry and provides a technical reference and theoretical guidance for development and application of novel bio-based films.

## 2. Materials and Methods

### 2.1. Materials

Bayberry (*Myrica rubra*) tannin powder (T, 85%) and eucalyptus (*Eucalyptus robusta Smith*) lignin (L, 98%) were purchased from Shengxuan Chemical Company (Zhengzhou, China). PVOH (molecular weight 250–300 × 10^3^ Da), glycerin, sodium hydroxide (30% aqueous solution), acetic acid (15% aqueous solution) were supplied from Sinopharm (Beijing, China), while *Penicillium* sp colonies were cultured from Wild Armillaria mellea (*Armillariellamellea* (*Vahl:Fr.*) *Karst*), collected in Zhaotong, China. It was put into sealed bags, moistened and cultured at room temperature for 24–72 h. After growth of mycelium on the surface of the bacteria, it was inoculated into sterile medium, and cultured at 28 °C and 75% humidity for 24–48 h to obtain *Penicillium* sp. colonies.

### 2.2. Preparation of Tannin/Lignin/PVOH (TLP) Film

Bayberry tannin was mixed with distilled water (DW) in a beaker under stirring for 5 min, then lignin and sodium hydroxide were added under stirring for 3 min, whereas glycerin was added and stirred for 3 min to obtain mixture 1. After that, polyvinyl alcohol was put into a three-necked flask, and dissolve at 90 °C, then it was mixed with mixture 1, and this was followed by addition of 15% glacial acetic acid to adjust the pH to 3, and the mixture was kept under stirring for 10 min to obtain TLP resin in liquid form. Afterwards, the TLP resin was put into a mold (10 × 10 × 1 cm^3^), which was left to dry spontaneously over 1 h to obtain TLP film. In addition, PVOH [23], lignin/PVOH (LP) film and tannin/PVOH (TP) film were prepared under the same conditions for comparison with TLP film. The raw materials and amounts are shown in Table 1. The preparation process of TLP film is illustrated in Figure 1.

### 2.3. Characterizations

#### 2.3.1. SEM Observation 

Scanning electron microscope (SEM S4800, Hitachi, Japan) was used to observe the cross-sectional morphology of the prepared film while operated at acceleration voltage of 20 kV. Before observation, the film was frozen in liquid nitrogen, then cut into a size of 5 mm × 5 mm × 4 mm using sharp scissors.

#### 2.3.2. FTIR and ESI-MS 

The structure of each film was investigated using a Varian 1000 infrared spectrometer (Varian, Palo Alto, CA, USA) and ESI-MS spectrometer (Waters, Milford, MA, USA). In the case of FTIR examination, the samples preparation involved mixing 1 g of KBr with 0.01 g of each sample in powder form and the investigation run covered a wave number range of 500 to 4000 cm^−1^. For ESI-MS testing, the TLP resin was dissolved in chloroform at a concentration of 10 μg/mL, then it was injected into the ESI source plus ion trap mass spectrometer (Bruker Daltonics Inc., Billerica, MA, USA) through a syringe at a flow rate of 5 μL/s. The relevant spectra were recorded in a positive mode with ion energy of 0.3 eV and a scanning range of 0 Da to 2000 Da.

#### 2.3.3. Differential Scanning Calorimetry (DSC) and Thermal Gravimetric Analysis (TGA) 

Differential scanning calorimeter (DSC, 204 F1, Netzsch, Selb, Germany) and thermogravimetric analyzer (TGA 209 F3, Netzsch, Selb, Germany) were used to check the thermal properties of the films, in which the DSC runs proceeded using 1.3~2.8 mg of the resin under nitrogen atmosphere at a heating rate of 3 °C/min over the temperature range from 30 °C to 300 °C. In the case of TGA process, 5~10 mg of cured film was used while protected by nitrogen atmosphere over the temperature range 30 °C~800 °C whereas a heating rate of 15 °C/min was undertaken.

#### 2.3.4. UV-_VL_ Resistance of the Various Films

The film resistance against UV absorption was performed in the wavelength range 250~800 nm using UV-_VL_ spectrophotometer (A260, Mettler Toledo, Zurich, Switzerland), on film samples cut in advance into dimensions of 4 cm × 1 cm × 0.2 mm.

#### 2.3.5. Mechanical Characterization 

The tensile properties of the film were evaluated with a universal mechanical testing machine (INSTRON-4467, Boston, MA, USA) according to the national standard GB/T 1040.3-2006. The film was cut to a size of 160 mm × 20 mm × 0.2 mm, the test distance was set at 50 mm while the tension rate of the sample was adjusted to 50 mm/min at ambient temperature around 23 °C, and humidity of 45%. The evaluation was repeated 5 times for each sample and average value was calculated.

#### 2.3.6. Biodegradability Test

The degradation test of the TLP film was conducted according to relevant literatures [30,31,32,33]. *Penicillium* sp. colony was used to provoke film degradation, in which the sample of TLP film was put into the culture dish, the *Penicillium* sp. colony was inoculated, and the culture dish was sealed, then cultivated at 28 °C and 75% relative humidity, and any change in the film weight or integrity was recorded for comparison between before and after culturing.

## 3. Results and Discussion

Figure 1 shows the FTIR spectra of PVOH, TLP, TP and LP films. It can be seen from the figure that the absorption peak in the range of 3402~3236 cm^−1^ is related to the O-H stretching vibration of alcohol hydroxyl and phenol hydroxyl. The peaks at 1627~1648 cm^−1^ belong to aromatic ring skeleton of different chemical environments; 1402~1415 cm^−1^ refers to the-CH_2_ absorption peak; Moreover, the peak at 2843 cm^−1^ in PVOH refers to the-CHO absorption peak, which indicates that parts of PVOH has undergone isomerization reaction to form dilute aldehyde structure under alkaline condition, and the peak at 2843 cm^−1^ in PVOH has shifted to 2885 cm^−1^ in LP, to 2899 cm^−1^ in TP and to 2854 cm^−1^ in TLP respectively, Meanwhile, the C=C absorption peak does not appear in LP, and the peaks at 1032 cm^−1^ and 1042 cm^−1^ are indicative of C-O stretching vibration [16], Interestingly, there is a C-O absorption peak in TP (1050 cm^−1^). These results indicate that isomerized PVOH reacts with tannin and lignin under acidic conditions. It can be also seen from the figure that the 3425 cm^−1^ absorption ascribed to O-H stretching vibration of TLP, which is shifted relative to the O-H stretching vibration in case of TP and LP, indicating that the hydrogen bond formed by the interaction of hydroxyl groups in tannin and lignin.These results indicate that the chemical reaction between tannin and lignin in PVOH system leads to a change of the chemical environment. However, the FTIR spectrum revealed the absorption peaks corresponding to many functional groups overlapped, in addition to some impurities exist in the lignin and tannin powder, which makes it difficult to accurately prove the reaction of tannin, lignin and PVOH through FTIR technique. Therefore, ESI-MS was used to further prove the co-condensation of tannin, lignin and PVOH.

Figure 2 displays the ESI-MS spectrum of TLP film. The peaks at 1334 Da and 1650 Da are attributed to the different fractions produced by the condensation of tannin, lignin and isomerized PVOH. These structures, to some extent, reveal that the condensation reaction takes place between tannin and polyvinyl alcohol and between lignin and polyvinyl alcohol. The main reaction among them is shown in Figure 2a,b. In addition, the peak at 1969 Da indicates that the chemical products of a and b (Figure 2) are connected by hydrogen bonds. Because of the uneven distribution of electron cloud and unstable double bond due to the enrichment of electrons, the cross-linking of polymers formed by lignin or tannin and isomerized PVOH segments makes the electron cloud in the conjugated system tend to be average.

Figure 3 shows the appearance and SEM images of PVOH, TP, LP, and TLP films. After adding the condensable tannin on the basis of PVOH, the TP color becomes reddish brown, while the tannin and PVOH present a “sea-island” structure, which indicates that tannin can be blended with PVA. Tannin is distributed in PVOH (sea) as an island (marked by a yellow circle). However, the distribution of tannin in PVOH is uneven, and the shape and size of the island are quite different. At the same time, in the case of the LP sample, lignin and PVOH are bonded together, so it is not easy to see the sea-island structure, however, there is still uneven mixing (yellow circle mark). The roughness and unevenness of the LP films may be due to the non-homogeneous distribution of lignin after mixing with PVOH, which is less compatible [31]. The SEM image of the TLP shows that the surface of the film is relatively smooth and flat compared to the addition of T and L alone, but a few protrusions can still be observed, probably due to the small amount of tannin and lignin not participating in the reaction along with uneven mixing. These results indicate that adding tannin and lignin together into PVOH can make the blend system more uniform. Lignin has a certain adhesion property; thus, it can better function with tannin to form a uniform system distributed in PVOH [28].

DSC thermograms of the different films are shown in Figure 4. As the crystallinity in PVOH film is high [22], its melting point is about 195 °C, however, the melting point of the PVOH film is reduced after adding condensed tannin or lignin separately or combined. For TP or TLP, when lignin or tannin is blended with PVOH alone, physical blending mainly occurs. During the mixing process, the tannin or lignin species can cause destruction of the chemical structure of PVOH which reduces the melting point. When tannin and lignin are added simultaneously, the melting point of the TLP is also reduced from 195 °C to 174 °C. Under acidic environment, the hydroxyl groups between tannin and lignin molecules can also interact with the hydroxyl groups on PVOH. Meanwhile, the chemical reactions among tannin, lignin and PVOH proceed, which also destroys the original ordered structure of PVOH, and reduces the melting point of TLP. However, the melting point of TLP film is higher than that of TP and LP films. According to the results of FTIR and ESI-MS, lignin and tannin reacted with PVOH and agglutinate with PVOH by hydrogen bonds. 

Figure 5 shows the TG and DTG curves of the various films. The thermograms reveal that the mass loss of all samples can be divided into three stages. The range of the first stage is 50 °C–150 °C, in which the mass loss of these films is about 10%, which is due to the evaporation of entrapped water in the films. The second stage extended between 150 °C and 350 °C, where the mass loss of the films in this stage is the largest whereas the loss rate is the fastest. This is revealing to the decomposition of the polymer chains with a mass loss around 70%. The third stage occupied the range between 350 °C and 550 °C where the weight loss attained about 10%. Upon the temperature reached 700 °C, the residual mass loss of these films became tiny and basically remained unchanged. It is worth noting that, with the increase in temperature, the mass loss of PVOH sample is greater than that of other films beyond 400 °C, indicating that the heat resistance of PVOH films is improved after PVOH is blended with tannin or lignin. The heat resistance of TP is superior with respect to LP, indicating that tannin exposes better heat resistance than lignin, whereas, on the other hand the heat resistance of TLP is between that of TP and LP. This result shows that the reaction amongtannin, lignin and PVOH does not enhance the thermal properties of the system, so the chemical bonding between tannin, lignin and PVOH is easy to be destroyed at high temperature. However, the TLP film still has good thermal stability compared with PVOH film, when the temperature reached 800 °C, the residual weight in the case of the TLP film was still about 20%.

Figure 6 demonstrates the tensile strength and elongation at break of these films. It can be clearly seen that compared with PVOH, the addition of tannin or lignin in the system is conducive to improving the tensile strength of the film. In particular, the addition of tannin makes the tensile strength of TP (31.3 MPa) higher than that of LP (13.9 MPa) and TLP (23.3 MPa). This depends on the high hardness of the tannin resin [13,15]. However, the addition of tannin reduces the elongation at break of the TP film, which reached 120% compared with 167% for PVOH. Interestingly, the addition of lignin can effectively improve the elongation at break of the film, rendering the LP film acquire higher elongation at break with respect to PVOH and TP. The reason for this result is that the toughness of lignin resin is higher than that of tannin resin [15,31]. Importantly, the elongation at break of TLP (358.4%) was more than 2 times that of LP (170.8%), which can be corroborated by chemical crosslinking reactions involving PVOH, tannin and lignin as confirmed by FTIR and ESI-MS results. Such crosslinking reactions promoted the formation of a toughened network structure, which improved the elongation at break of the film.

Figure 7 illustrates the UV-_VL_ transmittance curves of the different films. As can be seen from the figure, PVOH film without condensed tannin or lignin have much higher transmittance, which indicates that their anti UV-_VL_ ability is weak. The UV-_VL_ transmittance of films incorporating lignin and/or condensed tannin is significantly lowered compared with PVOH alone. The transmittance of TP films is lower than that of LP within the wavelength range of 250~700 nm. It shows that the ability of condensed tannin to absorb UV-_VL_ is stronger than lignin at that range. It is worth noting that the TLP film prepared by adding condensed tannin and lignin at the same time exhibited better ability to absorb UV-_VL_ within the wavelength range of 250~700 nm, and its light transmittance is lower than LP, or even TP. These transmittance values of the modified films present a strong proof that the physical blending and chemical reaction occurred among lignin, tannin and PVOH, which caused the improvement of PVOH ability to resist UV-_VL_.

After adding tannin and lignin into the PVOH system, the biodegradability test was also carried out. It is known that *Penicillium* sp. is a common omnivorous fungus with high heat tolerance and low cell reproduction temperature, and it only grows on substrates containing organic matter. Such properties allow it to be employed in the degradation of biomass materials better than bacteria [32,33]. More importantly, *Penicillium* sp. also has certain degradation effects on polyphenolic compounds, such as lignin and tannin [34,35,36]. The biodegradation process of TLP film sample under the action of *Penicillium* sp. is presented in Figure 8, from which, it can be seen that on the first day, a few *Penicillium* sp. colonies were implanted on the surface of TLP film (Figure 8a). Three days later, it can be observed that *Penicillium* sp. colonies propagate on the surface of the sample, with a faster propagation speed. As time goes on, it can be observed that on the 7th and 12th days, *Penicillium* sp. colony gradually diffuses in the culture dish, and a large number of *Penicillium* sp. hyphae covers the surface of TLP. This indicates that the TLP is resourcefully rich of the carbon required for *Penicillium* sp. growth. Further ahead, *Penicillium* sp. splits into mycelium on the film, and the mycelium grows downward, destroying hydrogen bonds and carbon chains from TLP sample until it is completely decomposed into water and carbon dioxide. Figure 8b illustrates that the film begins to the mass loss after 3 days from the *Penicillium* sp. action and increases with the time. The above results indicate the biodegradability of TLP film.

## 4. Conclusions

Composite films based on polyvinyl alcohol can be prepared while containing components from agricultural and forestry wastes, such as tannin and lignin from bayberry bark, via a simple procedure at room temperature. The produced bio-based film is considered interesting due to its good appearance and absence of cracks or bubbles. The physical blending of myrica rubra tannin, lignin, and PVOH took place at the same time. The reasonable compatibility of tannin and lignin with PVOH is responsible for the uniformed dispersion in PVOH matrix. Compared with pristine PVOH film, the TLP film showed better heat resistance and opposition against the detrimental effect of the ultraviolet and visible light. In addition, TLP acquired excellent tensile strength and elongation at break compared with neat PVOH film due to the marked chemical interaction between PVOH as matrix and the phenolic additives. At the same time, TLP film acquired obvious biodegradability, which is attributed to its main components derived from natural resources. For the near future, based on its good development prospects, it is feasible to regard TLP film as a kind of biomass based film to replace petrochemical based films to further achieve the “double carbon” goal not only in China but also abroad.

## Data Availability

Not applicable.

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
