# Peer review of "Facile Fabrication of High-Performance Composite Films Comprising Polyvinyl Alcohol as Matrix and Phenolic Tree Extracts"

_polymers, 2023, doi:10.3390/polym15061424_

Round 1

Reviewer 1 Report (Previous Reviewer 2)

-The authors should use PVOH for polyvinyl alcohol. (PVA can also be polyvinyl acetate which can make it confusing to readers).

-Revise the method of UV-Vis absorption on Lines 113-115 (the author already adjusted the wavelength ranges in the result).

-Explain more the DSC results relate to the mobility of molecules or changing of intermolecular force in the film matrix.

-Lignin can also show antimicrobial activity which may affect the biodegradation test in this study, please discuss more.

Author Response

1.The authors should use PVOH for polyvinyl alcohol. (PVA can also be polyvinyl acetate which can make it confusing to readers).

Reply to reviewer: I have changed PVA to PVOH in the full text and marked it with yellow highlighting.

  1. Revise the method of UV-Vis absorption on Lines 113-115 (the author already adjusted the wavelength ranges in the result).

Reply to reviewer: I have modified the UV-Vis range in the method and marked it with yellow highlighting.

  1. Explain more the DSC results relate to the mobility of molecules or changing of intermolecular force in the film matrix.

Reply to reviewer: I tried to explain it and marked it with yellow highlighting.

  1. Lignin can also show antimicrobial activity which may affect the biodegradation test in this study, please discuss more.

Reply to reviewer: Although lignin has anti-microbial properties, there are also some microorganisms that specifically destroy lignin, including Penicillium sp. We have emphasized it in the article and marked it with yellow highlighting.

Reviewer 2 Report (Previous Reviewer 1)

The author should emphasize the novelty of this study and disucess the result in detail.

Author Response

  1. The author should emphasize the novelty of this study and disucess the result in detail.

Reply to reviewer: We highlighted the novelty of the article and emphasized them in the results and discussions and marked it with yellow highlighting.

  1. We notice that the main text part of the paper is a little short (3523

words). As required in our website, for articles, it's suggested to have

a minimum word count of 4000 words

Reply to Editor: We have added more content and descriptions to reach 4000 words.

3. We noticed that the self-citation in your manuscript is a bit high, 11/34, 32%. In general, the self-citation is not more than 10%, please try to cite more references of other researchers or replace some of your references during revision.

Reply to Editor: We have reduced the self-citation rate to 10%.

This manuscript is a resubmission of an earlier submission. The following is a list of the peer review reports and author responses from that submission.

Round 1

Reviewer 1 Report

This article is based on the PVA matrix and additives from the tree extract, tannin, and lignin. The authors developed the facile fabrication process and analysis results of the composites. The SEM, FTR, ESI-MS, TGA, DTG, tensile test, UV-Vis, and biodegradation properties were demonstrated. My comments are listed below:

1.  Justify novelty in Introduction and Discussion. The study has to be hypothesis-driven.

2. There are two conclusions in the text.

3. It is necessary to explain the influence and mechanism of tannin and lignin additives on the various properties of the prepared composites.

Reviewer 2 Report

Abstract

- Please use PVOH for polyvinyl alcohol because PVA can also mean polyvinyl acetate.

Introduction

- Line 29, wear resistance [7-9], do the authors want to talk about 'tear resistance' or not?

Materials and methods

- 2.2, TLP resin in Lines 80 and 81 mean the mixture of film-forming solution in liquid form or solid? Please clarify.

- 2.3, for SEM analysis, how can the author cut the films? Cutting with a sharp blade or scissors will give different images.

- For UV absorption, the author used wavelength at 400-800 nm which represents the Visible light range, not the Ultraviolet range (250-400 nm). Please explain and revise in results and discussion part.

- For the degradation test, why only Penicillium sp. was used as the representative of the microbes for biodegradation? In soil or water, biodegradation can happen by a variety of microorganisms. Please explain.

Results

- FTIR results are not clear. Did the author arrange the spectra of films in the wrong order? Why PVOH alone showed more peaks than PVOH with tannin or lignin or tannin+lignin? Adding with T or L or T+L should be compared to the PVOH film's spectrum. Please revise.

- SEM images, does the island of T mean the agglomeration or the clump of T? How can the author prove it? Please explain.

- The author explained that dark spots in LP film are uneven mixing, but the authors also mentioned bonding together. Please make sure they are no cracks or void spaces in the film. Why does TLP have no clumps or cracks? Please explain. 

- For DSC thermograms, crystallinity can be observed from the delta H of peaks, the changing of melting points normally represents the changing of intermolecular reactions or forces. Please revise and clarify. Are there any relationships with the FTIR results?

- Tensile strength, TP showed the highest TS, and when L was added (as TLP), it reduced TS of TP film. That means L reduced the crosslink or intermolecular force between T and PVOH. The authors should reconsider the explanation of these results again. 

- Biodegradation result, Lines 241-244, author explained that "the mycelium grows downward, absorbing polysaccharides ". What kind of polysaccharide that the fungi can absorb from the film? Please explain.

- 4. and 5. are conclusions?

- From all comments in the results, Conclusions should be revised.